# Post-traumatic stress, depression and anxiety following the jet set nightclub collapse: Evidence from a nationwide prospective study in the Dominican Republic

Zoilo Emilio García-Batista[1] , Kiero Guerra-Peña[1], Adriana Alvarez-Hernandez[1], Antonio Cano-Vindel[2], Luciana Moretti[3] and Leonardo Adrián Medrano[4]

[1]Pontificia Universidad Catolica Madre y Maestra, Dominican Republic; [2]Complutense University of Madrid, Spain; [3]Universidad Empresarial Siglo 21, Argentina and [4]Universidad Nacional de Cordoba, Argentina

## Research Article

**Keywords:**
post-traumatic stress disorder (PTSD); mass disaster; mental health surveillance; emotional suppression; community resilience

**Corresponding author:**
Zoilo Emilio García-Batista;
Email: zoiloegarcia@gmail.com

## Abstract

This study examined the psychological impact of the Jet Set nightclub collapse in Santo Domingo, Dominican Republic, on April 8, 2025. Through a comprehensive approach, the research aimed to assess emotional symptoms in the immediate aftermath of the disaster. A prospective cross-sectional design was applied with a purposive sample of 1,034 adults who completed an online survey between April 9 and 15. Standardized instruments were used to evaluate post-traumatic stress disorder (PTSD), depression, anxiety, perceived social support and emotion regulation strategies. The study had three main objectives: (a) to estimate the prevalence of clinically significant emotional symptoms; (b) to analyze symptom variation according to trauma exposure level (direct, intermediate or vicarious); and (c) to explore sociodemographic and psychological predictors through multiple regression models. Results showed prevalence rates of 14.1% for PTSD, 27.9% for depression and 21.7% for anxiety. Higher symptom severity was observed among participants with direct exposure. Emotion suppression was consistently associated with greater psychological distress, while perceived social support was a protective factor against depression and anxiety. Female gender and younger age also emerged as significant predictors. These findings highlight the importance of considering both individual and contextual factors in post-disaster mental health responses and provide regionally relevant evidence to inform culturally sensitive interventions.

## Impact statement

This study represents the first nationwide psychological assessment in the Dominican Republic conducted immediately after a large-scale public disaster: the collapse of the Jet Set nightclub roof in April 2025. Within the first week of the tragedy, over 1,000 Dominican citizens were surveyed to understand the emotional and mental health impact of the event. The findings provide crucial evidence that both natural and man-made disasters can generate widespread psychological consequences, even among individuals who were not directly involved in the event.

The study highlights significant prevalence rates of probable post-traumatic stress disorder, anxiety and depression among the general population. It also identifies key risk and protective factors, such as emotional suppression and perceived social support, offering guidance for future preventive strategies. The results show that psychological distress extends beyond immediate victims and affects large portions of the community, underscoring the need for public mental health preparedness.

This research makes a direct contribution to mental health policy in low- and middle-income countries by offering data-driven recommendations for disaster response systems. It provides a blueprint for rapid mental health assessment following mass trauma and proposes practical, scalable strategies for governments, nongovernmental organizations and community leaders.

At a broader level, this study contributes to global discussions on trauma, community resilience and public health. It reinforces the urgency of integrating mental health into disaster management and preparedness plans, particularly in resource-constrained settings. The publication aims to raise awareness among policymakers, healthcare providers and international organizations about the emotional toll of public tragedies and the critical importance of immediate, structured mental health responses.





## Introduction

On April 8, 2025, the city of Santo Domingo witnessed one of the most devastating tragedies in the recent history of the Dominican Republic: the collapse of the Jet Set nightclub roof during a

crowded event. The catastrophe resulted in more than 230 deaths and left hundreds injured, not only physically but also emotionally, affecting survivors, eyewitnesses, relatives of the victims and the broader community. Among those affected were public figures, which intensified media coverage and the social resonance of the event. Beyond the immediate consequences in terms of infrastructure and physical health, events of this magnitude constitute potential traumatic experiences with severe and long-lasting psychological effects.

A substantial body of evidence supports the link between exposure to mass disasters and the development of psychopathology, particularly post-traumatic stress disorder (PTSD), depression and anxiety disorders. Previous studies in similar contexts have shown that disasters, whether natural or caused by human error or structural failures, can lead to high rates of PTSD and debilitating emotional symptoms (Neria et al. 2007; Goldmann and Galea, 2014). Moreover, recent research highlights that indirect exposure, such as through media coverage or emotional connection with the victims, can also trigger significant symptoms in populations not directly involved (Pfefferbaum et al., 2014).

These clinical conditions not only compromise individual well-being and psychosocial functioning, but if not identified and treated promptly, they can become chronic and lead to functional disability, social isolation and deterioration in quality of life (Bonanno et al., 2010; North and Pfefferbaum, 2013). In this context, early and systematic assessment of the psychological impact following traumatic events is essential for designing effective mental health responses aimed at alleviating suffering and preventing the development of persistent disorders.

In addition to assessing the psychological impact and emotional symptoms associated with the traumatic event, it is essential to consider the factors that may modulate the onset, intensity and persistence of these symptoms. Perceived social support has been consistently identified as a key protective factor. Feeling accompanied, understood and validated by meaningful networks, such as family, friends or the community, not only mitigates the psychological impact of trauma but also acts as a buffer against the development of severe emotional disorders (Ozbay et al., 2007; Thoits, 2011). The perception of social support influences multiple levels of emotional processing, including the interpretation of the traumatic event, the ability to coherently narrate the experience and the willingness to seek professional help, all of which are fundamental aspects of psychological recovery (Lakey and Orehek, 2011).

Likewise, emotional regulation strategies play a decisive role in individual coping following a traumatic experience. According to Gross's (1998) influential process model of emotion regulation, not all methods are equally effective. Cognitive reappraisal, the reinterpretation of a situation's meaning to reduce its emotional impact, has been associated with better psychological adjustment, lower levels of PTSD and greater subjective well-being (Troy et al., 2010; Aldao et al., 2010). In contrast, emotional suppression, which involves inhibiting the expression of emotions after they have been generated, tends to be linked to greater psychological distress, heightened physiological activation and poorer long-term mental health outcomes (Moore et al., 2008).

Despite the abundance of international research documenting the psychological effects of disasters in general, there remains a notable scarcity of empirical studies focused on Caribbean contexts, particularly in the Dominican Republic. This gap in the scientific literature not only obscures the psychosocial impact of such events on historically underrepresented populations but also limits the capacity of mental health systems to design culturally relevant and evidence-based interventions. Most psychological response models for emergencies have been developed and validated in countries of the global North, posing significant challenges for effective adaptation to different social, economic and cultural realities (Wessells, 2009; Kohrt and Hruschka, 2010).

In particular, the lack of systematic data in the Caribbean hinders the planning of context-sensitive public policies and weakens the training of professionals equipped to respond to collective crisis situations. As Patel et al. (2018) point out, mental health research in low- and middle-income countries remains insufficient, despite these regions bearing a high burden of unmet psychological suffering. Expanding the empirical base in these settings would not only enhance global knowledge about the effects of collective trauma but also enable the development of more effective, sustainable and culturally aligned clinical and community responses.

In this context, the main objective of the present study was to analyze the psychological impact of the catastrophe that occurred at the Jet Set nightclub through a comprehensive approach. To that end, the study aimed to: (a) estimate the prevalence of clinically significant symptoms of PTSD, depression and anxiety in the evaluated population; (b) examine differences in emotional symptomatology based on the degree of exposure to the event, distinguishing between direct and indirect exposure; and (c) explore the role of sociodemographic variables, perceived social support and emotion regulation strategies in predicting these symptoms, to identify relevant risk and protective factors in contexts of collective trauma.

## Methodology

### Participants

The sample for the present study consisted of 1,034 adult individuals residing in various provinces of the Dominican Republic (Table 1). Data collection was conducted remotely through a digital questionnaire administered between April 9 and April 15, 2025, in the days following the traumatic event. A nonprobabilistic, purposive sampling method was used. All participants provided informed consent before beginning the questionnaire.

Regarding demographic characteristics, participants' ages ranged from 16 to 82 years ($M = 40.25$, standard deviation [SD] = 13.21), with an interquartile range between 30 and 49 years, indicating broad representation of young and middle-aged adults. In terms of gender identity, 69.5% identified as female, 29.2% as male and 1.3% identified with another gender or preferred not to answer.

As for educational level, the majority had completed tertiary or university education (58.7%), while 29.4% had completed secondary education and 11.9% held postgraduate degrees (master's or doctorate). Regarding marital status, 41.2% reported being single, 38.5% married or in a domestic partnership and the rest were separated, divorced or widowed. At the time of the assessment, 78.5% of the sample were actively employed, distributed across various occupations, including public and private sector employees, independent professionals and informal workers. Participants were from both urban and rural areas, with a higher concentration in Santo Domingo and Santiago.

Concerning connection to the traumatic event, 12% of participants reported having been present at the scene of the collapse, while 24.3% indicated knowing individuals who were directly affected (injured or deceased). Additionally, 4.2% reported the loss of a family member or loved one. The remaining 59.5% of the sample did not report direct exposure or personal connections to

**Table 1.** Sociodemographic and exposure characteristics of the sample (N = 1,034)

| Characteristics | n | % |
|---|---|---|
| **Age group (years)** | | |
| 16–29 | 233 | 22.5 |
| 30–39 | 278 | 26.9 |
| 40–49 | 274 | 26.5 |
| 50–59 | 155 | 15 |
| 60+ | 83 | 8 |
| **Gender identity** | | |
| Female | 719 | 69.5 |
| Male | 302 | 29.2 |
| Other/prefer not to answer | 13 | 1.3 |
| **Education level** | | |
| Secondary | 304 | 29.4 |
| Tertiary/university | 607 | 58.7 |
| Postgraduate (master's/doctorate) | 123 | 11.9 |
| **Marital status** | | |
| Single | 426 | 41.2 |
| Married/cohabiting | 398 | 38.5 |
| Separated/divorced/widowed | 210 | 20.3 |
| **Employment status** | | |
| Employed (public/private sector, self-employed, informal work) | 812 | 78.5 |
| Unemployed | 222 | 21.5 |
| **Event exposure** | | |
| Present at the scene of the collapse | 124 | 12 |
| Knew individuals affected (injured/deceased) | 251 | 24.3 |
| Loss of a family member or loved one | 43 | 4.2 |
| No direct connection (indirect/vicarious exposure) | 615 | 59.5 |

affected individuals, constituting a group with indirect or vicarious exposure, likely influenced by media coverage or the widespread emotional climate following the event. Only 1.3% reported having received professional psychological care in the days following the incident.

## Instruments

*PTSD Checklist for DSM-5* (*PCL-5*): The Spanish version of the PCL-5 developed by Weathers et al. (2013) was used. It consists of 20 items assessing PTSD symptoms according to the Diagnostic and Statistical Manual of Mental Disorders, Fifth Edition (DSM-5) criteria. Responses are rated on a Likert scale from 0 (*not at all*) to 4 (*extremely*), with a total score range from 0 to 80. In this study, a cutoff score of ≥33 was adopted, validated as a clinical threshold for the general population (Bovin et al., 2016). The scale has demonstrated high internal consistency, with Cronbach's alpha values ranging from 0.91 to 0.95.

*Patient Health Questionnaire-9* (*PHQ-9*): Developed by Kroenke et al. (2001), this nine-item scale assesses depressive

symptoms over the past 2 weeks. Items are rated from 0 (*not at all*) to 3 (*nearly every day*), with a total score ranging from 0 to 27. A cutoff of ≥10 was used to identify moderate to severe symptomatology. The PHQ-9 has shown excellent internal reliability ($\alpha \approx 0.89$) and good convergent validity with other depression measures.

*Generalized Anxiety Disorder-7* (*GAD-7*): This scale by Spitzer et al. (2006) assesses generalized anxiety symptoms through seven items rated from 0 to 3. Total scores range from 0 to 21, and a cutoff of ≥10 was applied. Previous studies have reported high internal consistency ($\alpha = 0.92$) and adequate clinical sensitivity for use in research and screening.

*Emotion Regulation Questionnaire* (*ERQ*): The ERQ by Gross and John (2003) evaluates the use of two emotion regulation strategies: cognitive reappraisal (six items) and emotional suppression (four items), using a Likert scale from 1 (*strongly disagree*) to 7 (*strongly agree*). Both subscales have demonstrated good reliability ($\alpha = 0.79$–$0.89$) and discriminant validity with respect to other affective constructs.

*Multidimensional Scale of Perceived Social Support* (*MSPSS*): An adapted version of the MSPSS (Zimet et al., 1988) for the local context was used. It consists of 12 items assessing perceived social support from family, friends and other significant sources. Responses are given on a 7-point scale. This scale has been widely used in cross-cultural contexts and has shown adequate psychometric properties ($\alpha > 0.85$).

*Sociodemographic questionnaire*: In addition to the psychometric scales, an ad hoc questionnaire was developed to collect relevant sociodemographic information. This included questions on age, gender, educational level, marital status, employment status, place of residence and type of connection to the traumatic event (direct presence, knowledge of victims, loss of family members, etc.). This information was used to characterize the sample and contextualize the psychological results.

### Procedure and statistical analysis

This study employed a nonexperimental, cross-sectional and correlational design. Data collection took place between April 9 and April 15, 2025, through an online questionnaire distributed via social media, institutional platforms and community groups. Participation was voluntary, anonymous and unpaid, and informed consent was explicitly obtained before accessing the instrument.

After the data collection phase, the dataset was cleaned, coded and analyzed using the Python (pandas, scipy and statsmodels) and R (tidyverse and psych) statistical packages. Descriptive analyses were conducted (frequencies, means and SDs), as well as prevalence analyses of emotional symptomatology using clinically established cutoff scores. Group comparisons were performed using independent-samples *t*-tests.

To assess the predictive effects of social support and emotion regulation strategies on emotional symptoms, multiple linear regression models with standardized coefficients were conducted. Additionally, moderation models were estimated, including interaction terms between psychological and sociodemographic variables. Significance was set at $p < 0.05$.

All procedures were revised by the Ethics Committee of the Pontificia Universidad Católica Madre y Maestra and were carried out in accordance with the ethical guidelines of the Declaration of Helsinki.

## Results

### *Prevalence of emotional symptoms*

The prevalence of clinically significant emotional symptoms was estimated using validated cutoff scores from the literature for each of the scales employed. Regarding PTSD, a total of 14.1% of participants scored 33 or higher on the PCL-5 scale, the recommended threshold for identifying probable PTSD cases in the general population (Bovin et al., 2016).

When analyzing the PTSD symptom profile by PCL-5 dimensions, a moderate presence of symptoms related to intrusion, such as memories, dreams and involuntary reactions associated with the traumatic event, was observed. The most prominent items were "Unwanted memories of the event" ($M = 1.49$, SD = 1.14) and "Intense distress when reminded" ($M = 1.30$, SD = 1.20), followed by "Disturbing dreams" ($M = 0.95$, SD = 1.09). Less intense, though still notable, were "Feeling as if the event were happening again" ($M = 0.86$, SD = 1.14) and "Intense physical reactions" ($M = 0.74$, SD = 1.10).

In the avoidance dimension, which assesses active efforts to avoid trauma-related thoughts, emotions or stimuli, the item "Avoiding thoughts or feelings" was among the most highly rated on the entire scale ($M = 1.56$, SD = 1.18), whereas "Avoiding places or people" had a lower average ($M = 0.86$, SD = 1.14).

The dimension of negative alterations in cognition and mood showed elevated scores on the items "Persistent negative emotions" ($M = 1.12$, SD = 1.20), "Loss of interest in activities" ($M = 0.98$, SD = 1.19) and "Negative beliefs" ($M = 0.79$, SD = 1.10). In contrast, lower averages were found for "Exaggerated guilt" ($M = 0.34$, SD = 0.75), "Memory difficulties" ($M = 0.25$, SD = 0.68) and "Inability to feel positive emotions" ($M = 0.72$, SD = 1.06).

Finally, the dimension of alterations in arousal and reactivity, which reflects physiological hyperarousal and behavioral dysregulation, was primarily represented by the items "Being overly alert or tense" ($M = 1.10$, SD = 1.21), "Trouble sleeping" ($M = 1.03$, SD = 1.22), "Irritability" ($M = 0.98$, SD = 1.19) and "Concentration problems" ($M = 0.79$, SD = 1.12). In contrast, "Exaggerated startle response" ($M = 0.58$, SD = 0.97) and especially "Self-destructive or risky behavior" ($M = 0.19$, SD = 0.60) were the least frequent, with the latter being the lowest scoring item across the entire scale (Figure 1).

Regarding depressive symptomatology (Figure 2), 27.9% of participants scored 10 or higher on the PHQ-9 scale, indicating the presence of symptoms consistent with moderate to severe depression (Kroenke et al., 2001). Descriptive analysis revealed that the most prevalent symptoms were fatigue or lack of energy (reported by 53.8% of respondents with scores ≥2), loss of interest or pleasure in activities (anhedonia) (47.6%) and persistent feelings of sadness or hopelessness (44.2%). These findings highlight a high frequency of affective and somatic manifestations characteristic of depression in the days following the traumatic event, even among individuals who were not directly exposed. In contrast, the least reported symptoms were suicidal ideation (5.7%) and psychomotor changes, such as restlessness or slowing down (8.1%).

Regarding generalized anxiety (Figure 2), 21.7% of participants scored 10 or higher on the GAD-7 scale, exceeding the clinical threshold proposed to identify probable cases of GAD (Spitzer et al., 2006). The most frequently reported symptoms were excessive worry about various situations (42.1% of participants with scores ≥2), difficulty relaxing (40.8%) and a constant feeling of nervousness or tension (39.2%). These results indicate a significant burden of anxiety symptoms, both cognitive and somatic, of mild-to-moderate intensity, consistent with the expected psychological effects in contexts of indirect exposure to collective traumatic

events. In contrast, the least frequently reported symptoms were intense fear that something terrible might happen (20.1%) and physical restlessness or inability to remain calm (24.3%).

### *Emotional symptomatology and trauma exposure*

Levels of emotional symptomatology (PTSD, depression and anxiety) were assessed in relation to the degree of exposure to the traumatic event. Three distinct groups were established: direct exposure, which included individuals who were victims or present at the scene of the collapse; intermediate exposure (witness/connection), which comprised those who knew or lost affected individuals, or witnessed the event; and no direct contact, consisting of participants whose experience was exclusively vicarious, through media coverage.

Results from a one-way analysis of variance indicated statistically significant differences among the groups in average levels of PTSD, $F(2, N) = 9.56$, $p < 0.001$; depression, $F(2, N) = 6.07$, $p = 0.002$; and anxiety, $F(2, N) = 7.45$, $p < 0.001$. Post-hoc analyses using Tukey's honestly significant difference test revealed that the direct exposure group reported significantly higher symptom levels across all three dimensions compared to both the intermediate and no exposure groups ($p < 0.01$ in all cases). No significant differences were found between the intermediate exposure and the no-exposure groups.

Figure 3 illustrates these findings, showing a clear trend of increasing levels of emotional symptomatology as a function of trauma exposure. These results support the existence of a dose–response relationship between exposure to mass traumatic events and the development of emotional symptoms.

### *Risk and protective factors for emotional symptoms*

Multiple linear regression analyses were conducted to examine predictors of emotional symptoms following the catastrophe, using PTSD (PCL-5), depression (PHQ-9) and anxiety (GAD-7) scores as dependent variables. All models included perceived social support, emotion regulation strategies (cognitive reappraisal and emotional suppression) and sociodemographic variables (age, sex, marital status, employment status and educational level) as predictors. Reported coefficients are standardized ($\beta$).

For PTSD (Table 2), the model was significant, $F(8, 973) = 6.75$, $p < 0.001$ and explained 4.5% of the variance (adjusted $R^2 = 0.045$). Sex was a significant predictor ($\beta = -0.10$, $p = 0.002$), indicating higher symptom levels among women. A significant negative association was also found with age ($\beta = -0.11$, $p = 0.001$), suggesting that younger participants exhibited more severe symptoms. Emotional suppression showed a marginal trend toward significance ($\beta = 0.13$, $p = 0.056$), while cognitive reappraisal ($\beta = 0.03$, $p = 0.626$) and perceived social support ($\beta = -0.03$, $p = 0.348$) were not significant predictors. Marital status, employment status and educational level also showed no significant effects.

Regarding depression (Table 3), the model was statistically significant, $F(8, 936) = 10.38$, $p < 0.001$, with an adjusted $R^2$ of 0.074. Emotional suppression was positively associated with depression levels ($\beta = 0.18$, $p = 0.010$), while perceived social support showed a protective effect ($\beta = -0.09$, $p = 0.009$). Once again, sex ($\beta = -0.11$, $p = 0.001$) and age ($\beta = -0.17$, $p < 0.001$) were significant predictors, with higher symptom levels observed in women and younger participants. Cognitive reappraisal ($\beta = 0.00$, $p = 0.985$) and other sociodemographic variables were not significant predictors.

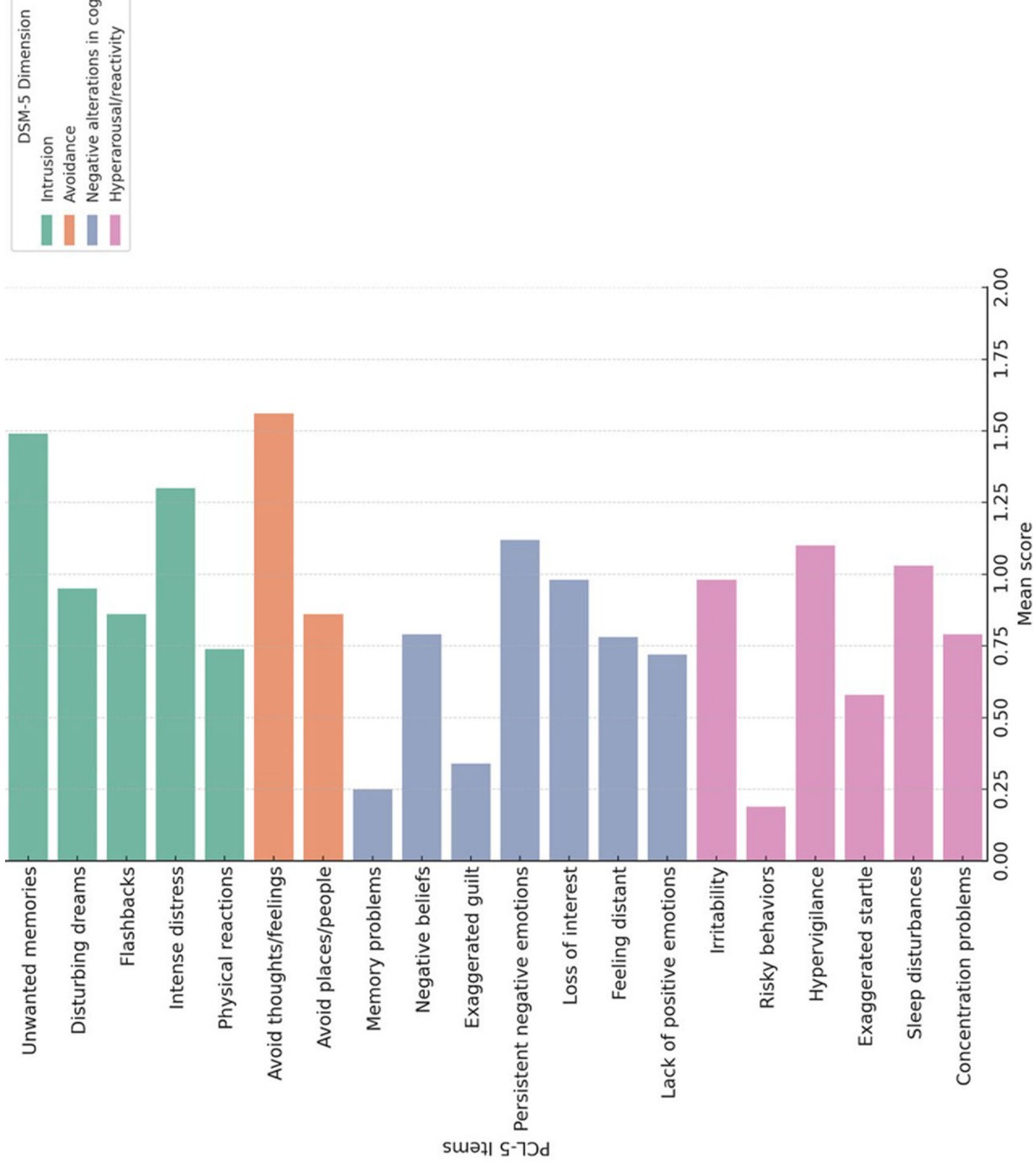

**Figure 1.** Average PTSD symptom scores by item (PCL-5), grouped by DSM-5 dimension.

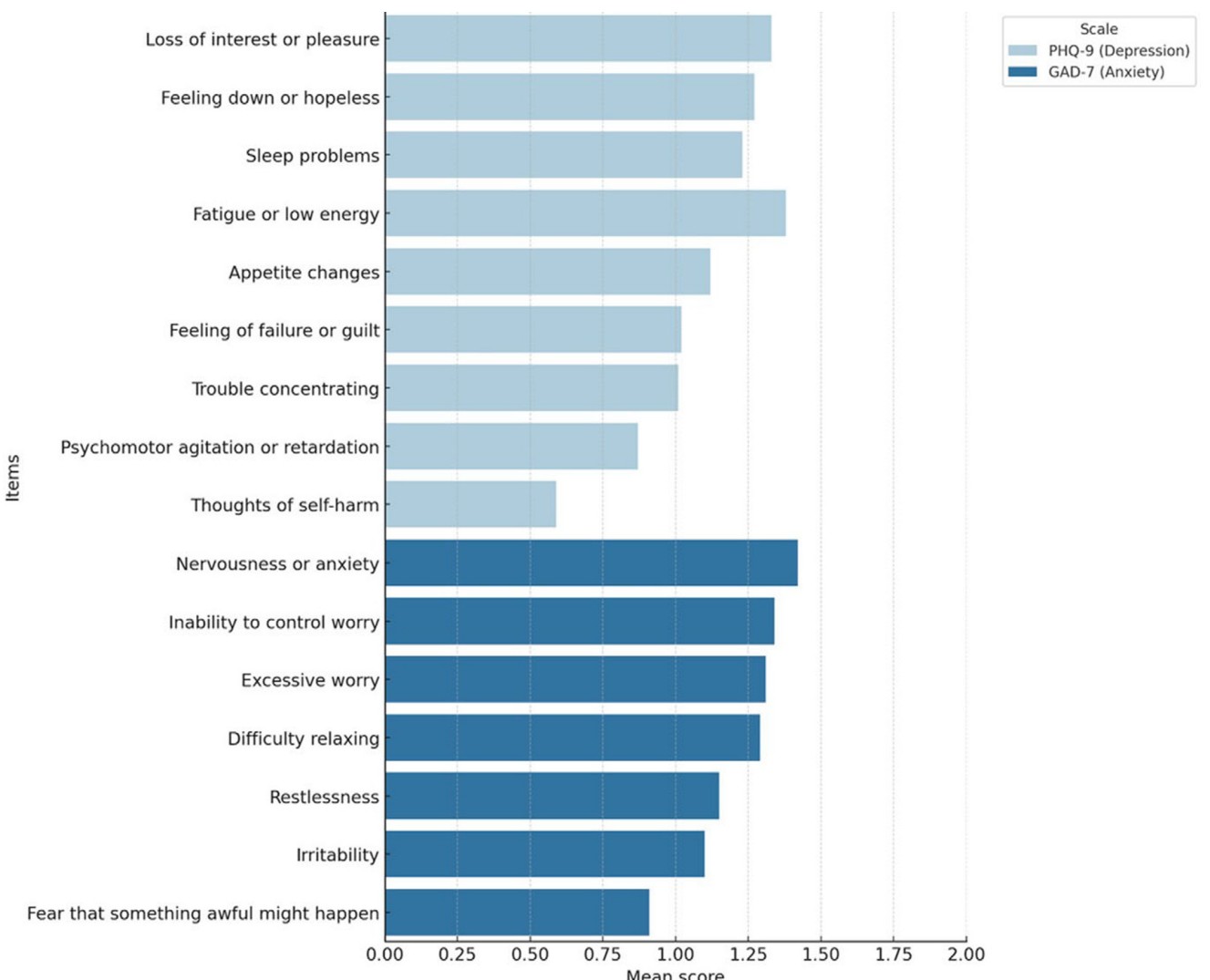

**Figure 2.** Average symptom scores by item: PHQ-9 (depression) and GAD-7 (anxiety).

The model for anxiety (Table 4) was also statistically significant ($F(8, 939) = 12.97$, $p < 0.001$), and explained 9.2% of the variance (adjusted $R^2 = 0.092$). Emotional suppression was a significant predictor ($\beta = 0.19$, $p = 0.005$), as was perceived social support, which was negatively associated with symptoms ($\beta = -0.11$, $p = 0.001$). Sex ($\beta = -0.14$, $p < 0.001$), age ($\beta = -0.16$, $p < 0.001$) and employment status ($\beta = -0.07$, $p = 0.025$) were also significant predictors, indicating higher levels of anxiety among women, younger individuals and unemployed participants. In contrast, cognitive reappraisal ($\beta = -0.01$, $p = 0.886$), marital status and educational level did not show significant effects in this model.

Taken together, these findings highlight the differential roles of emotional and sociodemographic variables in predicting psychological distress following a traumatic event, emphasizing in particular the clinical relevance of emotional suppression and perceived social support in levels of depression and anxiety symptoms.

## Discussion

The primary objective of this study was to analyze the psychological impact of the catastrophe that occurred at the Jet Set nightclub, addressing three key dimensions: the prevalence of clinically significant emotional symptoms, the relationship between symptomatology and degree of exposure to the event and the moderating role of psychological and sociodemographic variables. Regarding the first objective, the results revealed a considerable prevalence of emotional symptoms within the evaluated sample, with 14.1% of participants scoring in the range consistent with PTSD, 27.9% showing signs of moderate to severe depression and 21.7% reporting clinical symptoms of generalized anxiety. These figures are consistent with previous studies that report elevated rates of psychopathology in the weeks following mass disasters, even among individuals without direct exposure (Neria et al., 2007; Goldmann and Galea, 2014).

The dimensional analysis of PTSD showed a high presence of intrusion and avoidance symptoms, moderate scores on cognitive and affective alterations and lower, though still notable, levels of physiological hyperarousal. These findings support the hypothesis that exposure to high-impact traumatic events can trigger a broad spectrum of symptoms, affecting multiple areas of psychological functioning. Furthermore, the fact that depressive and anxious symptoms were also frequent among participants without direct exposure suggests the presence of emotional contagion or vicarious

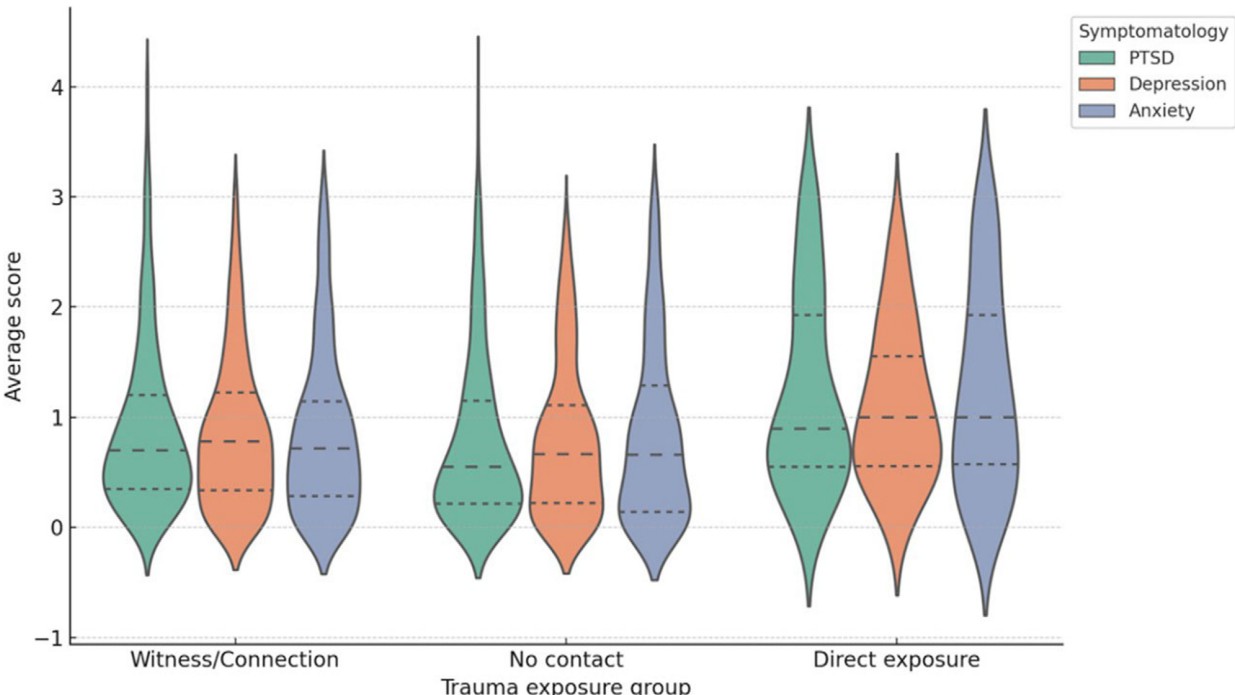

**Figure 3.** Emotional symptoms by trauma exposure group.

**Table 2.** Multiple regression predicting PTSD symptoms

| Predictor | B (95% CI) | SE | β | t | p |
|---|---|---|---|---|---|
| Constant | 19.24 (13.52, 24.96) | 2.92 | — | 6.60 | <0.001 |
| Cognitive reappraisal | 0.32 (−0.97, 1.61) | 0.66 | 0.03 | 0.49 | 0.626 |
| Emotional suppression | 1.23 (−0.03, 2.49) | 0.64 | 0.13 | 1.92 | 0.056 |
| Perceived social support | −0.38 (−1.18, 0.42) | 0.41 | −0.03 | −0.94 | 0.348 |
| Sex | −3.69 (−5.97, −1.41) | 1.16 | −0.10 | −3.18 | 0.002** |
| Age | −0.13 (−0.22, −0.05) | 0.04 | −0.11 | −3.20 | 0.001** |
| Employment status | −1.07 (−3.85, 1.71) | 1.42 | −0.03 | −0.75 | 0.451 |
| Marital status | 0.33 (−0.57, 1.22) | 0.46 | 0.02 | 0.71 | 0.476 |
| Educational level | −0.16 (−1.52, 1.21) | 0.69 | −0.01 | −0.23 | 0.822 |

*Note*: **$p < 0.01$.

**Table 3.** Multiple regression predicting depressive symptoms

| Predictor | B (95% CI) | SE | β | t | p |
|---|---|---|---|---|---|
| Constant | 11.07 (8.68, 13.47) | 1.22 | — | 9.07 | <0.001 |
| Cognitive reappraisal | 0.01 (−0.53, 0.54) | 0.27 | 0.00 | 0.02 | 0.985 |
| Emotional suppression | 0.69 (0.16, 1.21) | 0.27 | 0.18 | 2.56 | 0.010* |
| Perceived social support | −0.45 (−0.78, −0.11) | 0.17 | −0.09 | −2.63 | 0.009** |
| Sex | −1.65 (−2.59, −0.71) | 0.48 | −0.11 | −3.45 | 0.001** |
| Age | −0.08 (−0.12, −0.05) | 0.02 | −0.17 | −4.68 | <0.001** |
| Employment status | −0.65 (−1.81, 0.51) | 0.59 | −0.04 | −1.10 | 0.270 |
| Marital status | 0.11 (−0.26, 0.56) | 0.19 | 0.02 | 0.58 | 0.560 |
| Educational level | −0.20 (−0.77, 0.37) | 0.29 | −0.02 | −0.70 | 0.485 |

*Note*: *$p < 0.05$; **$p < 0.01$.

**Table 4.** Multiple regression predicting anxiety symptoms (*N* = 982)

| Predictor | *B* (95% CI) | SE *B* | *β* | *t* | *p* |
|---|---|---|---|---|---|
| Constant | 10.22 (8.15, 12.30) | 1.06 | — | 9.65 | <0.001 |
| Cognitive reappraisal | −0.03 (−0.50, 0.43) | 0.24 | −0.01 | −0.14 | 0.886 |
| Emotional suppression | 0.66 (0.20, 1.12) | 0.23 | 0.19 | 2.82 | 0.005** |
| Perceived social support | −0.48 (−0.78, −0.19) | 0.15 | −0.11 | −3.26 | 0.001** |
| Sex | −1.82 (−2.65, −0.99) | 0.42 | −0.14 | −4.31 | <0.001** |
| Age | −0.07 (−0.10, −0.04) | 0.02 | −0.16 | −4.68 | <0.001** |
| Employment status | −1.16 (−2.16, −0.15) | 0.51 | −0.07 | −2.25 | 0.025* |
| Marital status | 0.20 (−0.13, 0.53) | 0.17 | 0.04 | 1.17 | 0.241 |
| Educational level | −0.20 (−0.70, 0.30) | 0.25 | −0.03 | −0.79 | 0.433 |

*Note*: *p* < 0.05; **p* < 0.01.

impact mechanisms, possibly amplified by intense media coverage and the social resonance of the event (Pfefferbaum et al., 2014).

With respect to the second objective, statistically significant differences were found in levels of PTSD, depression and anxiety according to the degree of exposure to the traumatic event. Participants with direct exposure (those present at the site of the collapse) exhibited significantly higher levels of symptoms compared to those in the intermediate exposure group (individuals connected to victims) and the vicarious exposure group. These findings are consistent with the well-documented dose–response model in the trauma literature, which posits that greater proximity or intensity of exposure to a traumatic event is associated with a higher risk of developing psychopathology (Galea et al., 2005; North and Pfefferbaum, 2013).

Regarding the third objective, the multiple regression analyses identified significant predictors of emotional symptomatology. Emotional suppression was consistently associated with higher levels of PTSD, depression and anxiety, aligning with prior research documenting its maladaptive effects in coping with stressful events (Gross and John, 2003; Aldao et al., 2010). In contrast, perceived social support showed a protective effect against depressive and anxiety symptoms, reinforcing its role as an emotional buffer in crisis contexts (Ozbay et al., 2007; Thoits, 2011).

Contrary to what has been reported in numerous previous studies, cognitive reappraisal did not emerge in this study as a significant predictor of lower emotional symptomatology following the catastrophe. This strategy, conceptualized by Gross (1998) as a process of reinterpreting the meaning of a situation to modify its emotional impact, has been shown in various studies to be effective in reducing anxiety, depression and PTSD (Troy et al., 2010; Aldao et al., 2010). However, several factors may explain its lack of effectiveness in the context examined here.

First, the timing of the event must be considered. Data collection was conducted within the first week following the tragedy, a period characterized by emergency responses, physiological hyperactivation and fragmented emotional processing (Bonanno et al., 2010). During this stage, individuals tend to experience symptoms of intrusion, hypervigilance or emotional numbing, which can hinder the spontaneous and effective use of more complex cognitive strategies, including reappraisal.

Another relevant aspect is the sociocultural context. The effectiveness of emotion regulation strategies varies across cultures and prevailing coping styles. In community-oriented contexts like the Caribbean, where emotions are often expressed socially and coping tends to be more interpersonal, social support may play a more central and immediate role than individual cognitive regulation (Butler et al., 2007; Matsumoto et al., 2008). In this sense, cognitive reappraisal may not have been the preferred or prioritized strategy for managing emotional distress in the immediate aftermath of the trauma.

Among sociodemographic variables, being female and younger were both associated with higher levels of emotional symptoms, replicating well-established patterns in the literature on post-traumatic psychological vulnerability (Tolin and Foa, 2006). In the case of anxiety, employment status also emerged as a relevant factor: unemployed participants reported greater distress, suggesting that economic stress may act as an aggravating factor in the emotional impact following a disaster.

According to Tolin and Foa (2006), women show a higher prevalence of post-traumatic disorders due to a combination of biological, social and psychological factors, including heightened sensitivity to threatening stimuli, a greater use of internalizing coping strategies and higher cumulative exposure to stressful life events. These are compounded by sociocultural factors, such as gender inequality, role overload and a lower perceived sense of self-efficacy in extreme situations (Olff, 2017).

Additionally, age was inversely related to emotional symptomatology, with younger participants reporting higher levels of distress. This finding may be partly attributed to the fact that young adults often have less life experience, less emotional stability and fewer psychological resources for coping with large-scale crises (Norris et al., 2002). Furthermore, recent studies suggest that youth may amplify the impact of traumatic events due to their developmental stage of identity formation, vulnerability in social support networks and overexposure to media content (Arnett, 2000; Garfin et al., 2020).

## Practical implications and future research directions

The findings of this study provide concrete guidelines to improve psychological preparedness, intervention and recovery following collective traumatic events, especially in urban Latin American contexts such as the Dominican Republic. The identification of high prevalence rates of clinically significant symptoms of PTSD, depression and anxiety within the first week after the event underscores the need to implement early assessment strategies and immediate mental health responses, even before fully developed disorders manifest. As noted by Bonanno et al. (2010) and North

and Pfefferbaum (2013), timely intervention during the acute phases of trauma is key to preventing the chronicity of emotional distress and avoiding long-term functional impairments.

In this framework, it is essential to consider not only direct survivors but also individuals with indirect or vicarious exposure, who may develop significant symptomatology due to media coverage, emotional connection to victims or the broader social shock (Pfefferbaum et al., 2014). This implies expanding inclusion criteria in psychosocial support systems and developing tiered care protocols based on level of exposure.

The protective role of perceived social support, particularly against depressive and anxiety symptoms, highlights the importance of strengthening community-based, relational and collective interventions. As Ozbay et al. (2007) and Thoits (2011) emphasize, meaningful social networks not only buffer emotional distress but also facilitate access to resources, emotional regulation and validation of the traumatic experience. Community-based interventions, such as support groups, collective listening spaces or neighborhood care networks, may be especially effective in contexts where access to mental health professionals is limited (Kaniasty and Norris, 2008).

Likewise, the study confirmed the maladaptive role of emotional suppression, which was associated with higher levels of symptomatology across all three emotional dimensions assessed. The literature has shown that chronic suppression of emotional expression is linked to greater physiological reactivity, lower emotional recovery efficacy and higher risk of affective disorders (Gross and John, 2003; Moore et al., 2008). As a result, post-disaster intervention programs should include components that train more adaptive emotion regulation skills, such as cognitive reappraisal, acceptance and emotional expression in safe contexts (Berking and Wupperman, 2012).

Moreover, the fact that women, young people and unemployed individuals showed higher levels of psychological distress highlights the importance of designing differentiated interventions based on sociodemographic risk profiles. Previous studies have documented that women tend to experience higher levels of anxiety and depression following traumatic events, partly due to hormonal, social and cultural factors (Tolin and Foa, 2006; Olff, 2017). Similarly, young adults face trauma at a critical stage of identity development, often with fewer internal and external resources (Arnett, 2000), while unemployment has been identified as a strong predictor of emotional distress due to financial strain and loss of structure and purpose in life (Paul and Moser, 2009).

In this regard, it is recommended that post-disaster psychosocial response plans include prioritization criteria for women, youth and economically vulnerable populations, combining brief clinical approaches with accessible, mobile and community-based psychosocial interventions (Hobfoll et al., 2007). These strategies not only expand coverage but also promote equity in access to emotional care.

Although this study provides valuable empirical evidence on the psychological impact of a mass catastrophe in the Dominican Republic, some methodological limitations should be acknowledged when interpreting the results. First, a cross-sectional design was used, which limits the ability to establish causal relationships between the variables assessed. While significant associations were found, it cannot be definitively stated that the analyzed factors preceded or caused the emotional symptoms. Longitudinal studies would allow for tracking symptom trajectories over time and evaluating mechanisms of change or psychological adaptation.

Additionally, although some relevant sociodemographic and psychological variables were included, other contextual factors that could have influenced participants' emotional responses were not considered, such as previous mental health history, media exposure, religiosity, community belonging or perceived justice regarding the event.

From these limitations, several future research directions emerge. First, longitudinal studies would be valuable to follow the emotional trajectories of affected individuals over the weeks or months after the trauma, identifying both early and late risk and recovery factors. Second, it is recommended to advance the cross-cultural validation of instruments, such as the PCL-5, PHQ-9, GAD-7 and ERQ, in Caribbean populations to ensure clinical sensitivity and contextual relevance. Future research could also explore the interaction of individual, social and cultural factors through more complex statistical models, such as structural equation modeling or multilevel analysis.

Beyond the limitations of this study, the findings support the need to institutionalize post-emergency mental health surveillance and monitoring systems capable of collecting real-time data, identifying risk hotspots and coordinating multisectoral responses with an evidence-based approach. As Patel et al. (2018) suggest, low- and middle-income countries urgently need to strengthen their local mental health response capacities, moving beyond reactive care models and promoting preventive, integrated and culturally relevant approaches.

**Open peer review.** To view the open peer review materials for this article, please visit http://doi.org/10.1017/gmh.2025.10097.

**Data availability statement.** Researchers interested in obtaining the data from this study can communicate with the corresponding author to request access.

**Author contribution.** Zoilo Emilio García-Batista: conceptualization, project administration, investigation, writing – original draft and supervision; Leonardo A. Medrano: methodology, validation and formal analysis; Kiero Guerra-Peña: writing – review and editing and data curation; Adriana Alvarez-Hernández: writing – review and editing and investigation; Luciana Moretti: English language editing and academic writing support.
Antonio Cano-Vindel: Supervision, conceptual advice and resources.

**Financial support.** This work was supported by the Fondo Nacional de Innovación y Desarrollo Científfco y Tecnológico (FONDOCYT) of the Dominican Republic (grant number: 2022-3A1-241). This work was also supported by Vice-Rectorate for Research and Innovation of Pontificia Universidad Catolica Madre y Maestra (PUCMM), Dominican Republic.

**Competing interests.** The authors declare none.

**Ethics statement.** The study was reviewed by the Comité de Bioética de la Facultad de Ciencias de la Salud (COBEFACS) of the PUCMM. COBEFACS granted a waiver of formal ethics committee approval, as the research met the criteria for exemption from full ethical review.

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
