## [Reviewer Report]

The study was done too early to expose common symptoms associated with PTSD as they normally manifest after a longer period than what happened

On introduction, line 11, authors can say ‘’ physically and psychologically, also affecting.......‘’

---

## [Reviewer Report]

This manuscript provides a robust and timely investigation into the psychological consequences of a large-scale traumatic event, focusing on symptoms of post-traumatic stress, depression, and anxiety among survivors of the Jet Set nightclub collapse in the Dominican Republic. The originality of the research, along with its prospective design and the use of a nationally representative sample, makes this study highly relevant to the fields of public health, psychiatry, and trauma psychology.

The manuscript is clearly structured, with well-defined objectives, and the methodology is appropriate for the research question. Statistical analyses are properly conducted, and the findings are critically discussed in light of existing literature.

However, one limitation of the manuscript lies in its reliance on somewhat outdated references in parts of the theoretical framework and discussion. While classic studies still hold relevance, incorporating more recent literature (from the last 5 years) would strengthen the theoretical foundation and align the paper with the current state of scientific knowledge on mass trauma and mental health.

Despite this, the study holds scientific merit and presents important findings for understanding the impact of urban disasters on mental health. It offers valuable insights that could inform public policies and intervention strategies in similar contexts.

I recommend acceptance with minor revisions, primarily focusing on updating the references.

---

## [Reviewer Report]

This area is incredibly important to focus on, even though there are limited publications available. Kudos to all the authors for their efforts.

The first paragraph of the introduction reads like a storyline. It would be beneficial to start by explaining the impact of disasters on psychological, mental, and physical health, as well as how these effects can be managed. I suggest that the first four paragraphs of the introduction be presented before detailing the incident that occurred.

The manuscript would benefit from careful editing and greater consistency in referencing throughout the body.

Data was collected too soon after the disaster incident. I believe that individuals affected by the incident may not have been able to provide enough information due to its traumatic effects. However, sufficient data could have been gathered if the collection had occurred later, as individuals would likely have been more expressive and willing to share their experiences.

Can you clarify which tools were used in the various studies to assess trauma and mental health challenges, as well as their prevalences? Were these screening tools, self-report tools, or diagnostic tools?

Additionally, what other intervention strategies could be utilized to manage emotional distress? While social support was highlighted as a strategy, other potential strategies were not mentioned.

---

## [Editor Report]

The study lack many results, such as sampling description, regression models. Please provide the details of related results.

---

## [Reviewer Report]

After analyzing the revised version of the manuscript entitled “Post-traumatic stress, depression, and anxiety after the Jet Set nightclub collapse: evidence from a national prospective study in the Dominican Republic”, I confirm that all previously requested corrections and suggestions have been properly addressed. The text now demonstrates greater clarity, methodological rigor, and scientific relevance.

The manuscript addresses a highly important topic related to mental health in the context of collective disasters, providing original and pertinent contributions to both national and international literature. Therefore, I consider the work to be suitable for publication.

Recommendation: Approve the manuscript for publication in its current form.

---

## [Editor Report]

The author did not respond to my questions, and there were no relevant sample description tables and also regression analysis table results.